# Thiamine and METTL14 in Diabetes Management with Intensive Insulin Therapy

**DOI:** 10.3390/biomedicines13040980

**Published:** 2025-04-17

**Authors:** Miaoguan Peng, Yingying Zhang, Xiaoshi Weng, Jianfeng Wu, Taizhen Luo, Yanmei Dong, Shiyun Wen, Naifeng Liang, Liangying Zhong, Yaojie Zhai, Yijuan Xie, Yingjun Xie, Yuyi Chen

**Affiliations:** 1Department of Endocrinology, Guangdong Provincial Key Laboratory of Major Obstetric Diseases, Guangdong Provincial Clinical Research Center for Obstetrics and Gynecology, Guangdong-Hong Kong-Macao Greater Bay Area Higher Education Joint Laboratory of Maternal-Fetal Medicine, The Third Affiliated Hospital, Guangzhou Medical University, Guangzhou 510150, China; pmg@mail2.sysu.edu.cn (M.P.); xieyijuan1@163.com (Y.X.); 2Key Laboratory of Neurogenetics and Channelopathies of Guangdong Province and the Ministry of Education of China, Guangzhou Medical University, Guangzhou 510170, China; 3Department of Endocrinology, The Affiliated Brain Hospital, Guangzhou Medical University, Guangzhou 510170, China; 4Department of Obstetrics and Gynecology, Center for Reproductive Medicine, Guangdong Provincial Key Laboratory of Major Obstetric Diseases, Guangdong Provincial Clinical Research Center for Obstetrics and Gynecology, Guangdong-Hong Kong-Macao Greater Bay Area Higher Education Joint Laboratory of Maternal-Fetal Medicine, The Third Affiliated Hospital, Guangzhou Medical University, Guangzhou 510150, China; zhangyy_0601@163.com; 5Department of Obstetrics, Guangdong Provincial Key Laboratory of Major Obstetric Diseases, Guangdong Provincial Clinical Research Center for Obstetrics and Gynecology, Guangdong-Hong Kong-Macao Greater Bay Area Higher Education Joint Laboratory of Maternal-Fetal Medicine, The Third Affiliated Hospital, Guangzhou Medical University, Guangzhou 510150, China; 15013006343@163.com (X.W.); ltz751007@163.com (T.L.); yanmeidong320@163.com (Y.D.); 6Department of Obstetrics and Gynecology, Guangdong Provincial Key Laboratory of Major Obstetric Diseases, Guangdong Provincial Clinical Research Center for Obstetrics and Gynecology, Guangdong-Hong Kong-Macao Greater Bay Area Higher Education Joint Laboratory of Maternal-Fetal Medicine, The Third Affiliated Hospital, Guangzhou Medical University, Guangzhou 510150, China; w13908400450@163.com; 7Department of Endocrinology, Guangdong Provincial Key Laboratory of Major Obstetric Diseases, Guangdong Provincial Clinical Research Center for Obstetrics and Gynecology, Guangdong-Hong Kong-Macao Greater Bay Area Higher Education Joint Laboratory of Maternal-Fetal Medicine, The Third Clinical College of Guangzhou Medical University, The Third Affiliated Hospital, Guangzhou Medical University, Guangzhou 510150, China; 17388637562@163.com (S.W.); gzmulnf@163.com (N.L.); kyrazhai06@gmail.com (Y.Z.); 8Department of Clinical Laboratory, First Affiliated Hospital of Sun Yat-Sen University, Guangzhou 510080, China; zhongly6@mail.sysu.edu.cn

**Keywords:** m6A, *METTL14*, *TPK1*, *IPMK*, *PIK3R1*, insulin, diabetes mellitus, thiamine

## Abstract

**Background/Objectives**: Epigenetic regulation plays a critical role in diabetes research, with N6-methyladenosine (m6A) modification emerging as a key factor in disease progression. METTL14, an essential epigenetic regulator, may influence the effects of thiamine on intensive insulin therapy in diabetic patients. **Methods**: Blood samples from twenty diabetic patients were collected before and after intensive insulin therapy for MeRIP-seq and RNA-seq analysis. Genes with m6A modifications and corresponding mRNAs were identified and functionally analyzed using Gene Ontology (GO) and KEGG pathway analysis. RT-qPCR was used to confirm the overexpression of METTL14, PIK3R1, TPK1, and IPMK, while METTL14 overexpression was further validated in THP1 cells. **Results**: GO analysis revealed a significant enrichment of overlapping genes in metabolic pathways. A reduction in m6A modification levels was observed post intensive insulin therapy, indicating METTL14’s involvement in regulating TPK1, IPMK, and PIK3R1 expression. TPK1 levels showed a positive correlation with thiamine levels. Clinical validation demonstrated that combining thiamine with insulin therapy significantly reduced glucose and triglyceride levels compared to insulin alone. **Conclusions**: Thiamine supplementation alongside intensive insulin therapy offers therapeutic potential by downregulating TPK1 expression and mitigating lipid-related complications in diabetic patients. These findings highlight the pivotal role of METTL14-mediated m6A modification in regulating key metabolic genes during diabetes treatment.

## 1. Introduction

Diabetes mellitus is a multifaceted chronic disorder characterized by insulin resistance or inadequate insulin secretion from pancreatic β-cells, leading to elevated blood glucose levels [1]. This chronic hyperglycemia is associated with the damage and deterioration of vital organs such as the kidneys, eyes, heart, nerves, and blood vessels [2,3]. A primary goal in diabetes management is to prevent both microvascular and macrovascular complications [4]. Insulin therapy is the cornerstone of diabetes treatment [5], with advancements such as insulin analogs and delivery pumps enhancing glycemic control [6,7]. Despite these improvements, the mechanisms underlying the efficacy of insulin in diabetes remain unclear.

Recent advancements in epigenetic RNA modifications have illuminated the post-transcriptional regulation of gene expression [8]. Beyond environmental factors, epigenetic and post-translational modifications are pivotal in the pathogenesis of diabetes and its complications [9]. N6-methyladenosine (m6A) is the most prevalent mRNA modification in mammals, influencing a wide array of cellular processes [10,11]. m6A modification is catalyzed by writers, erased by erasers, and read by readers [12], and its alterations have been linked to β-cell dysfunction and insulin resistance [13]. Changes in m6A levels are associated with diabetic retinopathy, the primary microvascular complication of diabetes, and are influenced by inflammation, oxidative stress, and angiogenesis [14]. Yang Y et al. suggested that reduced m6A levels in type 2 diabetes patients may contribute to the upregulation of methyltransferases [15], and De Jesus DF et al. demonstrated that METTL14 knockout mice exhibit early-onset diabetes due to decreased β-cell proliferation and insulin secretion [16]. However, the global m6A methylation profile in diabetes remains underexplored.

Diabetes is a chronic metabolic disorder characterized by hyperglycemia, resulting from either insufficient insulin secretion or impaired insulin action [17,18]. m6A modification has been shown to regulate glucose and lipid metabolism, as well as immune and inflammatory responses, in diabetes [19]. Building upon this background, this study aimed to investigate the changes in m6A modification in diabetic patients following intensive insulin therapy using MeRIP-seq and RNA-seq. Our research aims to enhance the understanding of m6A methylation within the diabetes transcriptome and may inform the development of novel therapeutic strategies targeting m6A-modified mRNA.

## 2. Materials and Methods

### 2.1. Collection of Clinical Samples

Eligible participants for this study were individuals with type 2 diabetes aged between 30 and 65 years with a body mass index (BMI) ranging from 18.5 to 40 kg/m^2^, as per the 2017 edition of the Clinical Guidelines for the Prevention and Treatment of Type 2 Diabetes Mellitus in the Elderly in China. Exclusion criteria were stringent, including severe infections, acute complications, such as hyperglycemic hyperosmolar state, diabetic ketoacidosis, trauma, severe malnutrition, severe hepatic or renal insufficiency (with aminotransferase levels exceeding three times the normal value or an estimated glomerular filtration rate (eGFR) greater than 60%), type 1 diabetes, recurrent hypoglycemia, hyperthyroidism, hypothyroidism, pregnancy, lactation, or pregnancy intentions.

A total of 116 patients with type 2 diabetes were included in this study. Firstly, RNA sequencing was performed on the blood samples of three patients before and after intensive treatment. Then, based on the results of RNA sequencing, we detected the expression of METTL14 in the blood samples of 20 diabetic patients before and after intensive insulin therapy. For the thiamine efficacy assessment, an additional ninety-three patients were recruited, with twenty-three completing the study in the thiamine group (Group A) and seventy in the control group (Group B).

### 2.2. Patient Characteristics

Patient Characteristics are presented in Table 1.

The study design was a randomized, non-inferiority trial. The insulin dosage was individualized based on patient history: for those not previously on insulin, the initial total daily dose (U) was calculated as body weight (kg) multiplied by 0.5 U. This base amount represented 50% of the total daily insulin requirement, with the remaining 1/3 administered before each of the three main meals. For patients already receiving insulin therapy, the initial total daily dose (U) was 100% of the pre-pump total. The base amount was again 50% of the total, with the remaining 1/3 distributed pre-meal.

Blood samples were selected as the primary biospecimen based on clinical accessibility and established protocols for diabetes biomarker research. PBMC-derived epigenetic profiles have been shown to reflect systemic metabolic alterations relevant to insulin therapy responses. Clinical samples were collected in two distinct phases: (1) to evaluate m6A modifications associated with intensive insulin therapy in diabetic individuals, and (2) to assess the efficacy of thiamine supplementation in conjunction with intensive insulin therapy. Twenty diabetic patients were enrolled from the Third Affiliated Hospital of Guangzhou Medical University and were divided into two groups: Pre-intensive insulin therapy (Pre) and Post-intensive insulin therapy (Post).

For the thiamine efficacy assessment, an additional ninety-three patients were recruited, with twenty-three completing the study in the thiamine combination group. These patients were allocated to Group A, which received intensive insulin therapy with various treatment plans (including insulin pump therapy with insulin lispro, combined with dagliprazine 10 mg once daily or metformin 500 mg twice daily, escalating to 1000 mg twice daily by the fifth day), and Group B (70 patients), which was similar to Group A but included thiamine supplementation (10 mg three times daily) throughout the intensive treatment period. Blood samples were collected from all participants before and after the initiation of intensive insulin therapy to facilitate subsequent experimental analyses.

All enrolled patients underwent intensive insulin therapy using an insulin pump, specifically utilizing the insulin analog lispro insulin. Humalog (Insulin Lispro Injection) complies with the Import Drug Registration Standard (No. JX20020092) and holds the Import Drug Small Package Registration Certificate (No. H20090735) and Import Drug Large Package Registration Certificate (No. H20090736). The product is manufactured by Lilly France S.A.S. (Neuilly-sur-Seine, France) and repackaged under the National Drug Approval Number J20100005 by Lilly Suzhou Pharmaceutical Co., Ltd. (Suzhou, China), which is responsible for its packaging. The mean total daily dose of insulin lispro was (42.0 ± 7.8) U in Group A and (45.0 ± 5.6) U in Group B, with no statistically significant difference between the groups (*p* > 0.05).

Informed consent was obtained from all participants, and the study protocol was approved by the Medical Ethics Committee of the Third Affiliated Hospital of Guangzhou Medical University.

### 2.3. MeRIP-Seq and RNA-Seq

MeRIP-seq was conducted on the blood samples of three diabetes mellitus patients before and after intensive insulin therapy. Initially, total RNA was extracted and purified, subsequent to which ribosomal RNA was eliminated. The RNA was then cleaved into approximately 100-nt fragments using an RNA cleavage reagent. A subset of approximately 1/10 of the RNA fragments was set aside as input controls for subsequent RNA sequencing. The remaining samples were mixed with protein A/G magnetic beads and an anti-m6A antibody cocktail, which was incubated at 4 °C overnight. The m6A antibody was subjected to proteinase K digestion, after which the methylated RNA was purified for subsequent MeRIP-seq analysis. Raw reads from MeRIP-seq were aligned to the GRCh38/hg38 reference genome using STAR (v2.7.9a) with default parameters. m6A-enriched regions were identified using MACS2 (v2.2.7.1; parameters: ‘--nomodel--extsize 100--q-value 0.05’). Genomic coordinates of m6A peaks were mapped to chromosome bands using the UCSC Genome Browser’s cytoband track (GRCh38). Chromosomal distribution density was computed as the number of m6A peaks per 5 Mb genomic bin, normalized by total mapped reads (RPKM).

### 2.4. Gene Ontology (GO) and Kyoto Encyclopedia of Genes and Genomes (KEGG) Analysis

Based on the GO database “http://geneontology.org/ (accessed on 6 June 2022)”, the obtained m6A modification genes and mRNAs were annotated with GO from Biological Process (BP), Molecular Function (MF), and Cellular Component (CC) to screen out the significance of GO function. Pathway annotation of obtained m6A modification genes and mRNAs was conducted based on the KEGG database “https://www.kegg.jp/ (accessed on 6 June 2022)”, so as to select the significant pathway term.

### 2.5. Cell Culture and Transfection

THP1 cells were cultured in RPMI 1640 medium (C14-11875-093, Gibco, Grand Island, NY, USA), which contained 10% fetal bovine serum (FBS001Lifeman, URO, Guangzhou Abbiotechnology, Inc., Guangzhou, China), 0.05 mM 2-mercaptoethanol (21985-023, Gibco, USA), and 1% penicillin/streptomycin (HY-K1006, MCE, Monmouth Junction, NJ, USA). The incubation was carried out at 37 °C, in a chamber filled with 5% CO_2_ and saturated humidity. To genetically amplify *METTL14*, the *METTL14* sequences were conjugated to the pEnCMV vector, with the coding DNA region of *METTL14* inserted into the Hind and APA sites. Subsequently, according to the established protocol, pEnCMV and PencMv-*METTL14* were transfected into THP1 cells using Lipofectamine 2000 (11668027, Thermo, Waltham, MA, USA). The transfected THP1 cells were subjected to further analysis after a 48-h incubation period.

### 2.6. Quantitative Real-Time PCR (RT-qPCR)

Total RNA was extracted from the blood using Trizol (15596026, Thermo, USA), and then reverse-transcribed into cDNA by HiFiScript cDNA Synthesis Kit (CW2569, ConWin, Shenzhen, China). HotStart™ 2X SYBR Green qPCR Master Mix (K1070, APEXBIO, Houston, TX, USA) was applied to determine gene levels on QuantStudio™ 1 using *GAPDH* as an internal reference. *METTL14*, *PIK3R1*, *TPK1*, and *IPMK* levels were calculated by the 2^−ΔΔCt^ method. The primer sequences are as follows: *METTL14*-F: GGGGTTGGACCTTGGAAGAG, *METTL14*-R: CCCATGAGGCAGTGTTCCTT; *PIK3R1*-F: TCTTGTCCGGGAGAGCAGTA, PIK3R1-R: AGCCATAGCCAGTTGCTGTT; *TPK1*-F: AGGGAAAGCACAGGTTGCAT, *TPK1*-R: TGTGAGGTTCCACTTGAGGC; *IPMK*-F: TCTGGAGCAAGACAATGGGT, *IPMK*-R: TTGGCAACCAGTGGGAAGAT; *GAPDH*-F: GAGTCAACGGATTTGGTCGT, *GAPDH*-R: GACAAGCTTCCCGTTCTCAG.

### 2.7. Western Blot

Total protein was obtained using RIPA (P0013B, Beyotime, Shanghai, China). After quantification by BCA kit (BL521A, Biosharp, Hefei, China), the total protein was isolated by SDS-PAGE and transferred to PVDF membrane. PVDF membranes were then transferred to 5% skim milk and closed, and mixed with *METTL14* (26158-1-AP, Proteintech, Rosemont, IL, USA) and *GAPDH* (60004-1-Ig, Proteintech, USA) at 4 °C overnight. The membranes were then incubated with HRP-conjugated Affinipure Goat Anti-Rabbit IgG (H+L) (SA00001-2, Proteintech, USA). Finally, the membrane was exposed to ECL chemiluminescence solution (K-12045-D50, Advansta, San Jose, CA, USA) and visualized with ChemiScope6100 (CLiNX, Shanghai, China).

### 2.8. m6A Dot Blot Assay

According to previously reported literature [20], an evaluation of the alterations in the m6A modification levels of total RNA was conducted. The initial step involved the isolation of total RNA, subsequent to which mRNA was extracted. The mRNA was subjected to denaturation at 95 °C and then incubated on ice. The RNA was subsequently transferred onto a fiber membrane, which was exposed to ultraviolet light. The membrane was washed with TBST (QN1236-VGM, Bio Lebo, Wuhan, China) and incubated with an anti-m6A antibody at 4 °C. Following the incubation of the secondary antibody, the membrane was washed with TBST and incubated with Western blotting Substrate (Biosharp) at room temperature. The resulting dots were observed using a microscope (OI-X6) (Guangzhou, China).

### 2.9. Statistical Analysis

Graphpad Prism 8.0 software was employed for statistical computation and analysis. Normality of continuous variables was assessed via the Shapiro–Wilk test (*n* < 50) and Kolmogorov–Smirnov test (*n* ≥ 50). All measures underwent log (x + 1) transformation to meet normality assumptions. Data measurements were represented as mean ± standard deviation. A *t*-test was conducted to compare the two groups. A *p*-value of less than 0.05 signified a statistically significant difference.

## 3. Results

### 3.1. Distribution of Total m6A Modification Sites in People with Diabetes

First, we performed MeRIP-seq. Figure 1A shows the distribution density of total m6A modification sites on the chromosomes of three people with diabetes. Chromosome 1 showed the highest density, followed by chromosome 2 and chromosome 21. Furthermore, m6A modification sites were predominantly distributed in protein-coding regions, followed by other regions (Figure 1B). This suggests a possible association between m6A modification and protein coding. In addition, m6A modification sites were predominantly distributed in the 5′UTR and CDS regions of mRNA (Figure 1C). This observation suggested that m6A modification may be involved in protein coding and alternative splicing. Figure 1D shows the number of modification sites with common up-regulated and down-regulated m6A modification changes before and after intensive insulin therapy. A total of 774 genes were up-regulated and 417 genes were down-regulated.

### 3.2. Analysis of Differential Genes in m6A Modification Sites Before and After Intensive Insulin Therapy

We then performed GO and KEGG analysis on the differentially expressed genes in m6A modification sites before and after intensive insulin therapy in people with diabetes. The GO function revealed that genes with altered m6A modification were in cell, binding, cellular process, and metabolic process, among others (Figure 2A). Furthermore, the KEGG pathway was mainly enriched in neuroactive ligand-receptor interaction, oxytocin signaling pathway, and cGMP-PKG signaling pathway (Figure 2B).

### 3.3. Analysis of Differential mRNAs in People with Diabetes Before and After Intensive Insulin Therapy

We then performed RNA-seq. Figure 3A shows the heat map of common differential mRNA expression changes in three people with diabetes before and after intensive insulin therapy. Among them, 774 genes were down-regulated, and 417 genes were up-regulated. The GO function indicated that the commonly differentially expressed mRNAs were mainly located in Cell, Cell Part, Intracellular Part, Intracellular Binding, and Cell Progress, among others (Figure 3B). The KEGG pathway was mainly enriched in metabolic pathways, pathways in cancer, spliceosome, ubiquitin-mediated proteolysis, and herpes simplex infection (Figure 3C).

### 3.4. Analysis of Intersection Genes of Differential Genes in m6A Modification Sites and Differential mRNAs

Next, we took the intersection of genes with changes in m6A modification and changes in mRNA and plotted the differential expression heatmap (Figure 4A). The intersected genes included CENPF, TPK1, ZNF594, RAD54L2, PIK3R1, POLR2B, ZNF644, IPMK, PROK2, LINC00547, MTHFS and TMTC2. Among them, four genes were down-regulated, and eight genes were up-regulated. The GO function showed that the overlapping genes were mainly located in cell, binding, cellular process, and metabolic process, among others (Figure 4B). The KEGG pathway was mainly enriched in metabolic pathways, phosphatidylinositol signaling system, Epstein–Barr virus infection, a carbon pool by folate, and RNA polymerase (Figure 4C).

### 3.5. Analysis of m6A Alteration-Related Genes in Cutover Genes Before and After Intensive Insulin Therapy

We then screened the differential genes in the m6A modification sites based on the RNA SEQ results. We found that METTL14, YTHDF3, HNRNPH3, and RBM4 were downregulated in the Post group compared to the Pre group (*p* < 0.05) (Figure 5A). We also examined METTL14 expression in blood samples from 20 people with diabetes before and after intensive insulin therapy. Compared to the Pre group, METTL14 expression was downregulated in the Post group, and the difference was significant (*p* < 0.05) (Figure 5B). In addition, the m6A dot blot assay showed that the level of m6A modification decreased in people with diabetes after intensive insulin therapy (*p* < 0.05) (Figure 5C).

### 3.6. METTL14 Regulates the Expression of TPK1, IPMK and PIK3R1

Finally, we examined the expression of PIK3R1, TPK1, and IPMK in blood samples from 20 people with diabetes before and after intensive insulin therapy. Compared to the Pre group, PIK3R1 and TPK1 expressions were downregulated in the Post group, and the differences were significant (Figure 6).

The addition of thiamine is beneficial in reducing blood lipids in people with diabetes during intensive insulin therapy.

Clinical samples (Group A and Group B) with blood samples were taken from the diabetic subjects before and after the administration of thiamine supplementation based on intensive insulin therapy can reduce blood lipids in people with diabetes and reduce the risk of vascular complications in people with diabetes to some extent (Figure 7 and Figure 8 and Table 2).

## 4. Discussion

In this study, we utilize high-throughput sequencing to uncover changes in m6A modification in diabetes mellitus by implementing intensive insulin therapy in people with diabetes. By employing MeRIP-seq, we identified 774 downregulated genes and 417 upregulated genes with prevalent alterations in m6A modification prior to and subsequent to intensive insulin therapy. Moreover, we performed GO and KEGG analyses to decode the potential function of differentially expressed genes occurring at m6A modification sites. Through the integration of MeRIP-seq and RNA-seq, we identified the intersection genes of differentially expressed genes in m6A modification sites and differential mRNAs in people with diabetes before and after intensive insulin therapy. Ultimately, we validated the efficacy of the intersection genes in vitro. Our study substantiates a reduction in the level of m6A modification in people with diabetes following intensive insulin therapy. Through further validation, we revealed that METTL14 controls the expression of TPK1, IPMK, and PIK3R1. Our investigations provide an extensive reference for future discussions on the molecular mechanism underlying METTL14 in diabetes.

The enzymatic writers WTAP, METTL3, METTL14, and KIAA1429, the erasers *FTO* and *ALKBH5*, and the m6A binding protein reader YTH domain dynamically and reversibly regulate the biological effects of m6A modification [21]. In essence, m6A modification is governed by m6A methyltransferase and demethylase, which dictate the fate of target mRNA by influencing splicing, translation, and decay [22]. Currently, m6A regulatory factors are anticipated to serve as biomarkers for enhancing diabetes mellitus management. Consequently, investigating differentially expressed genes in m6A modification sites may offer insights into diabetes treatment. In this study, we employed MeRIP-seq and RNA-seq to uncover that the intersection genes of differential genes in m6A modification sites and differential mRNAs predominantly contribute to metabolic pathways in people with diabetes before and after intensive insulin therapy. Edhager AV et al. demonstrated the progressive changes in major metabolic pathways through myocardial proteomics in rats during the development of type 2 diabetes [23]. Gardi N et al. reported that metabolic pathways could mitigate the adverse prognosis associated with pancreatic (head) adenocarcinoma complicated by diabetes [24]. These investigations underscore the crucial role of metabolism in diabetes. Through further validation, we observed a decline in m6A modification levels in people with diabetes following intensive insulin therapy. Consequently, we proceeded to explore the expression of intersection genes of differential genes in m6A modification sites and differential mRNAs in people with diabetes before and after intensive insulin therapy.

According to RNA-seq outcomes, distinctive genes in m6A modification sites were identified, revealing a downregulation of METTL14 expression following intensive insulin therapy. METTL14 has demonstrated its ability to control β-cell function and diabetes mellitus [25]. Furthermore, METTL14 appears to play a critical role in diabetic nephropathy through the m6A modification of α-klotho [26]. This suggests that METTL14 might serve as a biomarker for diabetes treatment. Consequently, we delved further into its functionalities. Research indicates that thiamine levels decline with decreasing TPK1, and thiamine supplementation can mitigate diabetes [27,28]. Our study also proved that thiamine supplementation on the basis of intensive insulin treatment can reduce the blood lipids of people with diabetes and reduce the risk of vascular complications in people with diabetes to a certain extent, but the lowering effect on blood sugar is not obvious. Considering that thiamine supplementation takes a short time, Therefore, the positive effect of thiamine on the blood sugar of people with diabetes cannot be denied, and longer and larger sample studies are needed to verify this point. Jung IR et al. uncovered that IPMK mediates insulin signaling and gluconeogenesis, which may represent a potential therapeutic target for diabetes [29]. *PI3K* serves as a crucial component of insulin action, and the PIK3R1-encoded regulatory subunit is essential for transmitting insulin signaling through *PI3K* [30]. Karadoğan AH et al. reported that PIK3R1 emerges as a significant candidate gene in type 2 diabetes development, playing a pivotal role in insulin signaling transduction [31]. These studies collectively imply that TPK1, IPMK, and PIK3R1 contribute significantly to insulin signaling transduction. In our study, we confirmed that METTL14 regulates the expression of TPK1, IPMK, and PIK3R1 in THP1 cells transfected with an METTL14 overexpression vector. This suggests that *METTL14* might influence insulin signaling transduction to enhance diabetes by regulating the expression of TPK1, IPMK, and PIK3R1. Nonetheless, further investigations into the underlying mechanisms are warranted.

Mechanistically, thiamine may exert its metabolic effects through (1) restoration of mitochondrial oxidative capacity (via PDH activation), (2) modulation of PPP-derived NADPH to balance lipid synthesis and oxidation, and (3) indirect regulation of epitranscriptomic markers (e.g., m6A) through metabolic intermediate-driven signaling. The selective impact on lipid metabolism over glucose control could reflect thiamine’s stronger influence on NADPH-dependent pathways critical for lipogenesis and redox homeostasis, whereas glucose regulation involves redundant hormonal and non-enzymatic mechanisms less sensitive to thiamine sufficiency in non-deficient states.

The selective reduction in triglycerides without significant glucose-lowering effects may reflect thiamine’s preferential influence on lipid metabolism via the pentose phosphate pathway. By enhancing transketolase activity and NADPH production, thiamine could redirect metabolic flux toward lipid oxidation and away from triglyceride synthesis. In contrast, glucose homeostasis in critically ill patients is governed by multifactorial mechanisms (e.g., insulin resistance, counterregulatory hormones), which may explain the limited observed impact of thiamine supplementation. Future studies should assess baseline thiamine status and NADPH/redox biomarkers to further elucidate this dichotomy.

Based on preclinical and clinical evidence, we propose that the interaction between intensive insulin therapy and thiamine supplementation may involve (1) synergistic enhancement of glucose metabolism, (2) attenuation of oxidative stress, (3) support of hepatic regeneration, and (4) mitigation of the risk of hypoglycemia [32]. These hypotheses are based on thiamine’s role as a cofactor for pyruvate dehydrogenase and transketolase, which are critical for bridging glycolysis, the Krebs cycle, and the pentose phosphate pathway. Further mechanistic and clinical studies are warranted to validate these interactions. The hypotheses are consistent with previous studies demonstrating the efficacy of thiamine in the benefits of IIT for glycemic control. We recognize the need for future research to test these mechanisms empirically, particularly in patient cohorts receiving combined therapies.

While this study provides novel insights into blood-based epigenetic markers, we acknowledge that diabetes pathogenesis involves complex tissue-specific interactions. The observed methylation changes in circulating cells may represent both cell-autonomous effects and systemic metabolic crosstalk, which will require validation in relevant metabolic tissues through future collaborative studies. Although THP1 was chosen for its genetic manipulability in mechanistic studies, we acknowledge its limitations in mimicking in vivo cell–matrix interactions. Future studies will use patient-derived iPSC macrophages to improve clinical fidelity.

## 5. Conclusions

We delved into the alterations in m6A modification sites following intensive insulin therapy in diabetes mellitus utilizing MeRIP-seq, RNA-seq, and in vitro cellular experiments. Our research corroborated a decline in m6A modification levels post-intensive insulin therapy among people with diabetes. Moreover, we discerned the intersectional gene as well as the enrichment pathways for differential genes in m6A modification sites and variant mRNA pre- and post-intensive insulin therapy. Through further validation, we established that METTL14 governs the expression of TPK1, IPMK, and PIK3R1. Furthermore, it was observed that the expression of *TPK1* exhibited a decrease following strengthening, and the supplementation of thiamine during the strengthening process was found to be beneficial for enhancing blood lipid levels in people with diabetes. The current findings should be interpreted in light of sample source limitations. Although blood-based analyses provide clinically actionable insights, parallel investigations in pancreatic and hepatic tissues would be necessary to establish direct pathogenic mechanisms. Our study offers invaluable insights into the role of m6A modification in diabetes mellitus and the involvement of thiamine in the treatment of diabetic complications, thereby unveiling novel therapeutic targets for the management of this disease.

## Figures and Tables

**Figure 1 biomedicines-13-00980-f001:**
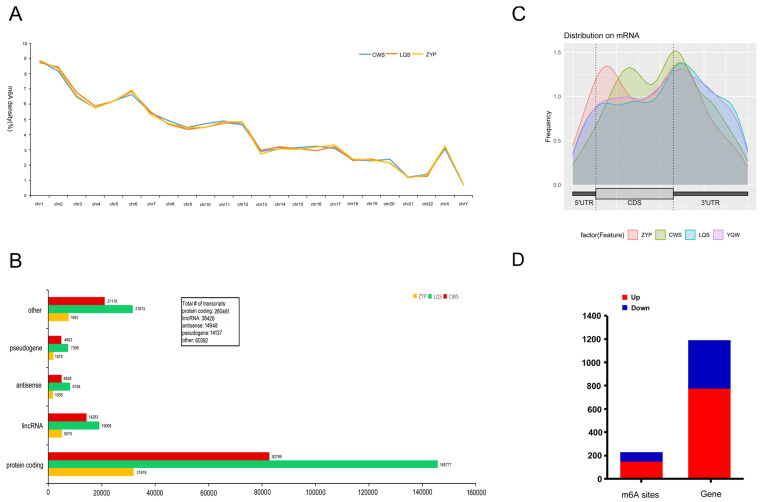
Total m6A modification sites in people with diabetes. (**A**) Distribution density of total m6A modification sites on chromosomes in 3 people with diabetes. (**B**) m6A modification sites were mainly distributed in protein-encoding regions, followed by lncRNA. (**C**) m6A modification sites were mainly distributed in the 5′UTR and CDS regions of mRNA. (**D**) Number of modification sites with common up-regulated and down-regulated m6A modification changes before and after intensive insulin therapy (*p* < 0.05). N6-methyladenosine, m6A; Long non-coding RNA, lncRNA; 5′ untranslated region, 5′UTR; coding sequences, CDS.

**Figure 2 biomedicines-13-00980-f002:**
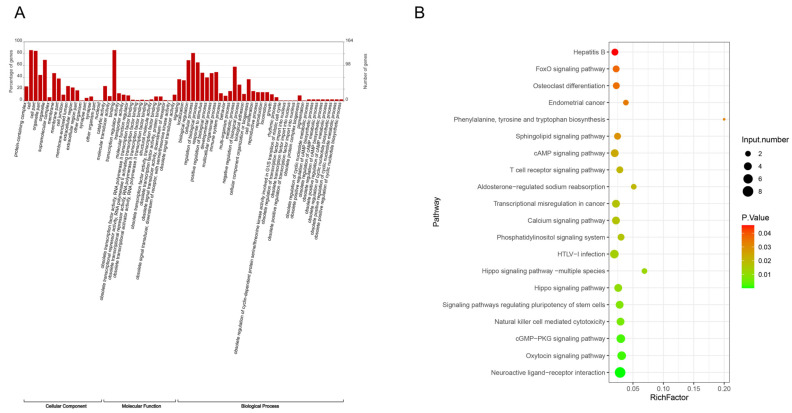
Analysis of differential genes in m6A modification sites in people with diabetes before and after intensive insulin therapy. (**A**) GO functional analysis of genes with altered m6A modifications. (**B**) KEGG enrichment pathway of genes with altered m6A modifications.

**Figure 3 biomedicines-13-00980-f003:**
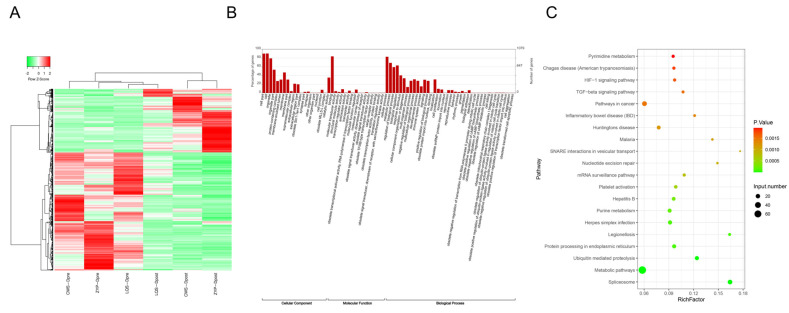
Analysis of differential mRNAs in people with diabetes before and after intensive insulin therapy. (**A**) Heatmap of common differential mRNA expression changes before and after intensive insulin therapy. (**B**) GO functional analysis of commonly differentially expressed mRNAs in three people with diabetes before and after intensive insulin therapy. (**C**) KEGG enrichment pathway of commonly differentially expressed mRNAs in three people with diabetes before and after intensive insulin therapy.

**Figure 4 biomedicines-13-00980-f004:**
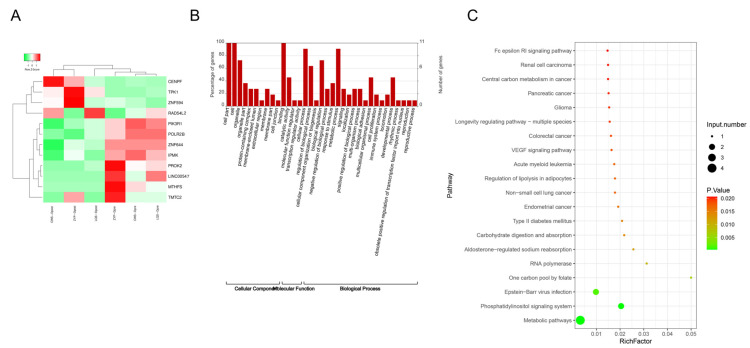
Analysis of intersection genes of differential genes in m6A modification sites and differential mRNAs. (**A**) Heatmap of intersection genes. (**B**) GO functional analysis of intersection genes. (**C**) KEGG enrichment pathway of intersection genes.

**Figure 5 biomedicines-13-00980-f005:**
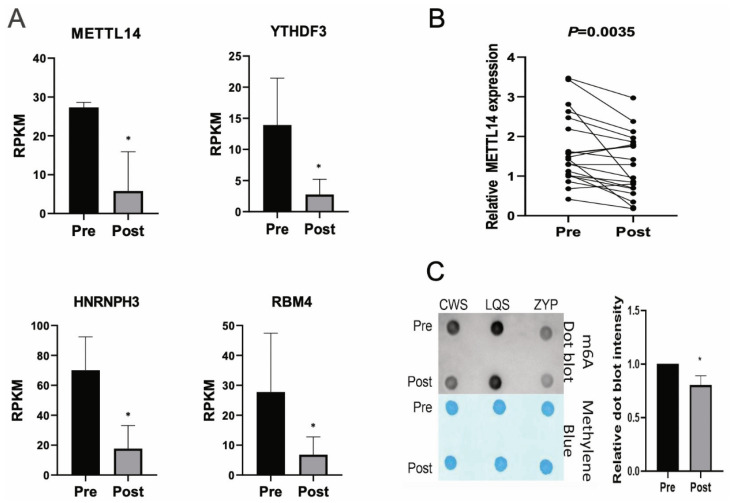
Analysis of m6A modification-related genes in intersection genes before and after intensive insulin therapy. (**A**) Differential genes in m6A modification sites were screened based on RNA-seq results. (**B**) Expression of METTL14 in blood samples of 20 people with diabetes before and after intensive insulin therapy was evaluated by RT-qPCR. (**C**) m6A dot blot assay was utilized to determine the m6A modification level in people with diabetes after intensive insulin therapy. * *p* < 0.05.

**Figure 6 biomedicines-13-00980-f006:**
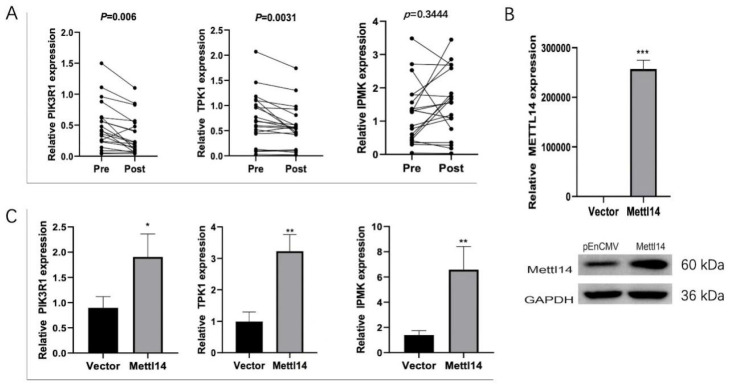
METTL14 regulated TPK1, IPMK, and PIK3R1 expressions. (**A**) PIK3R1, TPK1, and IPMK expressions were assessed by RT-qPCR in the blood samples of 20 people with diabetes before and after intensive insulin therapy. (**B**) Verification of the efficiency of overexpression of METTL14. (**C**) mRNA expression of TPK1, IPMK and PIK3R1 after METTL14 overexpression was measured by RT-qPCR. * *p* < 0.05, ** *p* < 0.01, and *** *p* < 0.001.

**Figure 7 biomedicines-13-00980-f007:**
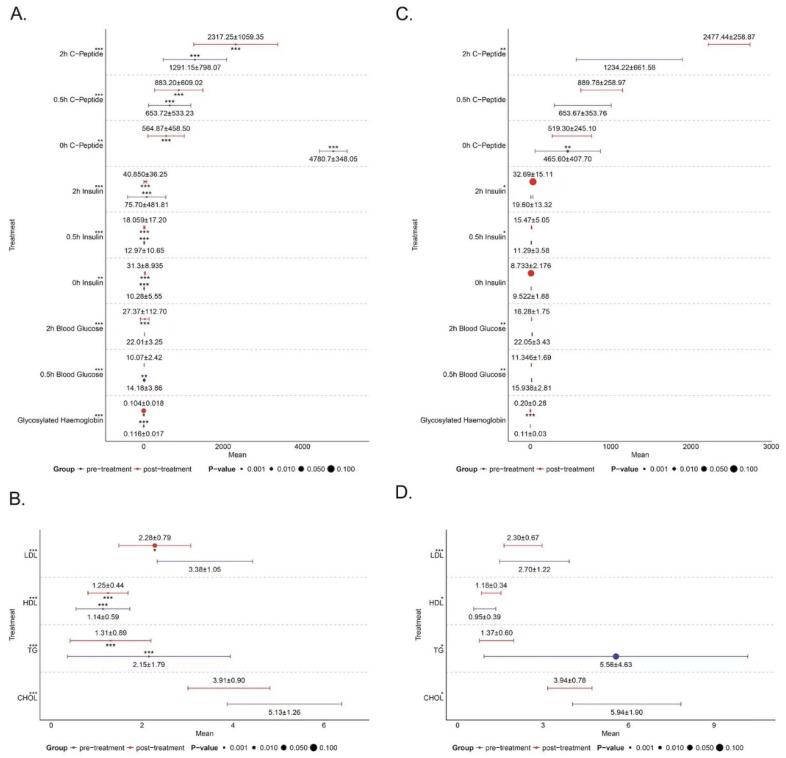
The addition of thiamine is beneficial in reducing blood lipids in people with diabetes during intensive insulin therapy. (**A**) Without the addition of thiamine, the blood glucose decreased after intensive insulin treatment, and the glycosylated hemoglobin and 2 h blood glucose showed significant differences before and after treatment. (**B**) Without thiamine, after intensive insulin treatment, the expression of blood lipids decreased, and the *p*-values of CHOL (total cholesterol), HDL (high-density cholesterol), LDL (low-density cholesterol), and TG (triglyceride) were all less than 0.05, indicating that they had significant differences before and after treatment. (**C**) With the addition of thiamine, the *p*-values of glycosylated hemoglobin and 2 h blood glucose were less than 0.05, indicating a significant difference before and after treatment. (**D**) With the addition of thiamine, the *p*-values for CHOL, HDL, and TG were all less than 0.05, indicating that they had significant differences before and after treatment. However, the *p*-value of LDL was greater than 0.05, indicating that there was no significant difference between LDL-low-density cholesterol before and after treatment. * *p* < 0.05, ** *p* < 0.01, *** *p* < 0.001.

**Figure 8 biomedicines-13-00980-f008:**
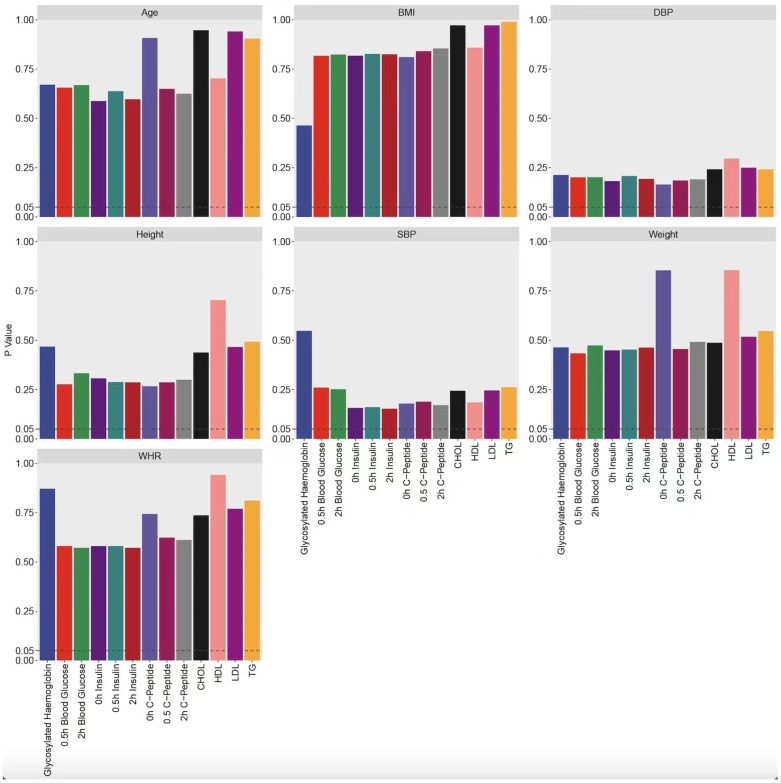
The addition of thiamine is beneficial in reducing blood lipids in people with diabetes during intensive insulin therapy. Analysis of personal data (age, height, weight, BMI, WHRDBP, SBP) showed no significant baseline differences between treatment groups A and B.

**Table 1 biomedicines-13-00980-t001:** Baseline characteristics and effects of short-term intensive insulin therapy treatment in two intervention groups, based on whether they took thiamine or not. Values are means (standard deviation) unless stated otherwise.

Characteristic	Three Diabetic Patients	Twenty Diabetic Patients	Twenty-Three Diabetic Patients with Thiamine (Group A)	Seventy Diabetic Patients Without Thiamine as Control (Group B)	Group A vs. Group B	
					*t* or χ^2^	*p*-Value
Age, years	53.67 (7.51)	51.4 (11.32)	55.76 (8.93)	54.49 (8.29)	4.57	0.79
Gender *:						
Male	2	12	15	46	0.00	0.97
Female	1	8	8	24		
Estimated disease duration, months	6.33 (2.89)	4.47 (4.60)	10.36 (3.95)	10.10 (4.33)	0.69	0.49
Systolic blood pressure, mm Hg	135.33 (9.61)	137.05 (17.79)	132.54 (20.31)	132.85 (13.70)	−0.95	0.34
Diastolic blood pressure, mm Hg	92.00 (10.39)	86.80 (10.12)	88.15 (12.38)	87.76 (11.47)	1.28	0.20
Body mass index	29.67 (9.23)	24.84 (2.99)	25.5 (3.78)	25.46 (4.06)	1.51	0.13
Waist circumference, cm	90.00 (3.61)	92.10 (7.98)	93.23 (8.89)	93.29 (6.98)	−2.71	0.79
Waist-to-hip ratio	0.92 (0.03)	0.96 (0.06)	0.97 (0.06)	0.98 (0.07)	−0.92	0.36

* Sexual data were presented as constituent ratios and analyzed using the chi-square test. Other data were assessed for normality and confirmed to follow a normal distribution, followed by analysis via independent sample *t*-tests.

**Table 2 biomedicines-13-00980-t002:** The addition of thiamine resulted in a significant decrease in TG-triglyceride levels. The difference reflects changes in values before and after intensive treatment. The group designated as Group A functioned as the pre-thiamine treatment cohort, and a comparative analysis was conducted between Group B and Group A subsequent to undergoing intensive therapy. The reduction in triglycerides in the group receiving intensive insulin treatment supplemented with thiamine was significantly greater compared to those receiving only intensive insulin treatment.

Indicator	*n*	Normal Test		Detection of Difference Significance	
		*p*-Value (Pre-Treatment)	*p*-Value (Difference)	*p*-Value (Post-Treatment)	*p*-Value (Difference)
glycosylated hemoglobin	82	0.000	0.000	0.285	0.306
2 h blood glucose	96	0.067	0.000	0.972	0.890
0.5 h blood glucose	95	0.006	0.038	0.126	0.625
0 h insulin	97	0.000	0.001	0.950	0.490
0.5 h insulin	95	0.000	0.000	0.844	0.746
2 h insulin	97	0.000	0.000	0.995	0.490
0 h C-Peptide	100	0.000	0.000	0.411	0.805
0.5 h C-Peptide	97	0.000	0.000	0.551	0.886
2 h C-Peptide	100	0.000	0.001	0.928	0.339
CHOL	91	0.291	0.422	0.108	0.097
TG	91	0.000	0.000	0.034	0.003
HDL	88	0.000	0.000	0.278	0.274
LDL	92	0.200	0.309	0.103	0.133

Note: Difference = each group (post-treatment value and pre-treatment value).

## Data Availability

Data that support the findings of this study are available from the corresponding author upon reasonable request.

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
