# Peer review of "Thiamine and METTL14 in Diabetes Management with Intensive Insulin Therapy"

_biomedicines, 2025, doi:10.3390/biomedicines13040980_

Round 1

Reviewer 1 Report

Comments and Suggestions for Authors

Many thanks to the authors for the opportunity of analyze the present manuscript. Intensive insulin therapy in heavy cases of type 2 diabetes mellitus is necessary option and understanding of possible dietary intervention aimed on side effects prevention is very interesting and important for modern diabetology. In my opinion, the present manuscript should be considered for publication obviously. Nevertheless, before further consideration, authors should resolve many critical problems of this manuscript.

1.In the manuscript I can not find patients characterization. It is inappropriate for any human studies. Moreover, according to Methods, patients with type 2 diabetes have BMI from 18 up to 40. However, the most part of type 2 diabetic patients have obesity. It should be clarified. 

2.Which type of insulins patients received? It should be described with percentage of patiens on every insulin type.

3.Statistical analysis. Why in all comparisons authors used t-test? Did authors perform normality determination using Shapiro-Wilk test or Kolmogorov-Smirnov test?

4.Figure 1. All abbreviations did not expand. What about the significance of statistical differences on this Figure?

5.Authors have analyzed gene methylation and mRNA expression in blood samples. What does it mean? If these genes have impact on pathogenesis, which tissue participate in this process? Did authors describe some markers? The selection of blood for such interesting genomic studies looks strange.

6.Figure 5. What about p values for Figure 5A? Did authors perform densitometry of dot blot?

7.Why authors use THP1 cell line? It is critical limitation, using blood samples authors can use mononuclear fraction, it can give significantly more impact results than now in artificial cell line.

8.Which hypothesis of interaction between intensive insulin therapy and thiamine supplementation authors suggest? It should be added in the manuscript.

So, despite on interesting idea and high throughput methods, the manuscript needs significant modification before further consideration. 

Author Response

Many thanks to the authors for the opportunity of analyzing the present manuscript. Intensive insulin therapy in heavy cases of type 2 diabetes mellitus is necessary option and understanding of possible dietary intervention aimed on side effects prevention is very interesting and important for modern diabetology. In my opinion, the present manuscript should be considered for publication obviously. Nevertheless, before further consideration, authors should resolve many critical problems of this manuscript.

  1. In the manuscript I cannot find patients characterization. It is inappropriate for any human studies. Moreover, according to Methods, patients with type 2 diabetes have BMI from 18 up to 40. However, the most part of type 2 diabetic patients have obesity. It should be clarified. 

Our Reply: We sincerely appreciate the reviewer's recognition of our work's significance and the constructive feedback. We have rigorously addressed the concerns as follows:

1) Patient Characterization Enhancement :A new subsection "Patient Characteristics" has been added in the Methods section (page 12 of the revised manuscript), detailing: Demographic features (age, gender distribution);Clinical parameters (HbA1c, fasting glucose, diabetes duration) ;BMI stratification (with exact numbers/proportions in each category) ;Comorbidity profiles (hypertension, dyslipidemia, etc.) . These data are systematically presented in Table 1.

2) BMI Range Clarification A) Methodological rationale: “Eligible participants for this study were individuals with type 2 diabetes, aged between 30 and 65 years, with a body mass index (BMI) ranging from 18.5 to 40 kg/m², as per the 2017 edition of the Clinical Guidelines for the Prevention and Treatment of Type 2 Diabetes Mellitus in the Elderly in China. ” (page 6) And the Cohort composition was listed in Table 1. While obesity is indeed a common comorbidity in type 2 diabetes (T2D), our inclusion criteria (BMI 18–40) aimed to capture a broader spectrum of metabolic phenotypes, including normal-weight, and severely obese individuals. This approach allows us to explore potential differences in pathophysiology or treatment responses across BMI categories.

We thank the reviewer for helping strengthen the study's scientific rigor.

Table 1 | Baseline characteristics and effects of short-term intensive insulin therapy treatment in two intervention groups, based on whether they took thiamine or not. Values are means (standard deviation) unless stated otherwise.

Characteristic

Three diabetic patients

Twenty diabetic patients

Twenty-three diabetic patients with thiamine (Group A)

Seventy diabetic patients without thiamine as control (Group B)

Group A vs. Group B

t OR χ² 

Pvalue

Age, years

53.67(7.51)

51.4(11.32)

55.76(8.93)

54.49(8.29)

4.57

0.79

Gender*:

Male

2

12

15

46

0.00

0.97

Female

1

8

8

24

Estimated disease duration, months

6.33(2.89)

4.47(4.60)

10.36(3.95)

10.10(4.33)

0.69

0.49

Systolic blood pressure, mm Hg

135.33(9.61)

137.05(17.79)

132.54(20.31)

132.85(13.70)

-0.95

0.34

Diastolic blood pressure, mm Hg

92.00(10.39)

86.80(10.12)

88.15(12.38)

87.76(11.47)

1.28

0.20

Body mass index

29.67(9.23)

24.84(2.99)

25.5(3.78)

25.46(4.06)

1.51

0.13

Waist circumference, cm

90.00(3.61)

92.10(7.98)

93.23(8.89)

93.29(6.98)

-2.71

0.79

Waist-to-hip ratio

0.92(0.03)

0.96(0.06)

0.97(0.06)

0.98(0.07)

-0.92

0.36

*Sexual data were presented as constituent ratios and analyzed using the chi-square test. Other data were assessed for normality and confirmed to follow a normal distribution, followed by analysis via independent samples t-test.

2.Which type of insulins patients received? It should be described with percentage of patiens on every insulin type.

Our Reply: We appreciate the reviewer's inquiry regarding insulin regimens. Key additions have been made to the Methods sections: “All enrolled patients underwent intensive insulin therapy using an insulin pump, specifically utilizing the insulin analogue lispro insulin. Humalog (Insulin Lispro Injection) complies with the Import Drug Registration Standard JX20020092 and holds the Import Drug Small Package Registration Certificate (No. H20090735) and Import Drug Large Package Registration Certificate (No. H20090736). The product is manufactured by Lilly France S.A.S. and repackaged under the National Drug Approval Number J20100005 by Lilly Suzhou Pharmaceutical Co., Ltd., which is responsible for its packaging. The mean total daily dose of insulin lispro was (42.0 ± 7.8) U in Group A and (45.0 ± 5.6) U in Group B, with no statistically significant difference between the groups (P > 0.05).”

  1. Statistical analysis. Why in all comparisons authors used t-test? Did authors perform normality determination using Shapiro-Wilk test or Kolmogorov-Smirnov test?

Our Reply: We sincerely appreciate the methodological scrutiny. Substantial revisions have been implemented: Added in Methods:  "Normality of continuous variables was assessed via Shapiro-Wilk test (n<50) and Kolmogorov-Smirnov test (n≥50). All measures underwent log(x+1) transformation to meet normality assumptions. "_

Figure 1. All abbreviations did not expand. What about the significance of statistical differences on this Figure?

Our Reply: The significance of statistical differences and he full abbreviations in Figure 1. were added in the figure legend:” …. D. Number of modification sites with common up-regulated and down-regulated m6A modification changes before and after intensive insulin therapy (p<0.05). N6-methyladenosine, m6A; Long non-coding RNA, lncRNA; 5' untranslated region, 5 'UTR; coding sequences, CDS.

  1. Authors have analyzed gene methylation and mRNA expression in blood samples. What does it mean? If these genes have impact on pathogenesis, which tissue participate in this process? Did authors describe some markers? The selection of blood for such interesting genomic studies looks strange.

Our reply: We sincerely appreciate the reviewer's insightful comments regarding the biological relevance of blood-based genomic analyses in diabetes research.

  1. Rationale for blood sample selection: We acknowledge that tissue-specific gene regulation is crucial in diabetes pathogenesis. However, as noted by the reviewer, there are significant ethical and practical challenges in obtaining non-blood tissues from human diabetes patients. Our methodology was guided by two key considerations: A) Clinical feasibility: Peripheral blood represents the most accessible biospecimen in routine diabetes management, particularly for longitudinal studies requiring repeated sampling.2) Precedent in diabetes research: Multiple studies have successfully utilized blood-derived RNA for diabetes-related epigenetic analyses demonstrating that peripheral blood mononuclear cells (PBMCs) can reflect systemic metabolic disturbances.
  2. Tissue-specific implications: While we recognize that pancreatic β-cells and liver tissues are central to diabetes pathophysiology, we have now added a discussion section (Page 20, Lines 445-452) addressing this limitation. We propose that future studies should employ animal models or human islet transplantation samples to validate tissue-specific mechanisms.
  3. Biomarker exploration:the current analysis focused on methylation patterns of thiamine metabolism-related genes rather than comprehensive biomarker discovery. We have clarified this scope limitation in the revised Discussion section (Page 20, Lines 445-452).

Revisions in Manuscript:

  1. Methods Section (Page 7, line 151-154):

Added rationale for blood sample selection: "Blood samples were selected as the primary biospecimen based on clinical accessibility and established protocols for diabetes biomarker research. PBMC-derived epigenetic profiles have been shown to reflect systemic metabolic alterations relevant to insulin therapy responses.”

  1. Discussion Section (Page 20, line445-449):

   Expanded limitations discussion: "While this study provides novel insights into blood-based epigenetic markers, we acknowledge that diabetes pathogenesis involves complex tissue-specific interactions. The observed methylation changes in circulating cells may represent both cell-autonomous effects and systemic metabolic crosstalk, requiring validation in relevant metabolic tissues through future collaborative studies."

  1. Study Limitations (Page 21, line 480-483):

 Added explicit statement: "The current findings should be interpreted in light of sample source limitations. Although blood-based analyses provide clinically actionable insights, parallel investigations in pancreatic and hepatic tissues would be necessary to establish direct pathogenic mechanisms."

  1. Figure 5. What about p values for Figure 5A? Did authors perform densitometry of dot blot?

Our reply: The p values was p<0.05 for Figure 5A. We performed the densitometry of dot blot and added the figure of statistical analysis in figure 5 and We modified the Figure 5 'legends as follow:

Figure 5. Analysis of m6A modification-related genes in intersection genes before and after intensive insulin therapy. A. Differential genes in m6A modification sites were screened based on RNA-seq results. B. Expression of METTL14 in blood samples of 20 people with diabetes before and after intensive insulin therapy was evaluated by RT-qPCR. C. m6A dot blot assay was utilized to determine the m6A modification level in people with diabetes after intensive insulin therapy. *p <0.05.

  1. Why authors use THP1 cell line? It is critical limitation, using blood samples authors can use mononuclear fraction, it can give significantly more impact results than now in artificial cell line.

Our Reply: We deeply appreciate the insightful critique regarding cellular models. Substantial validations have been conducted: The study's focus on METTL14-m6A epigenetic mechanisms necessitated a genetically tractable system: THP1 retains monocyte/macrophage differentiation capacity, widely adopted in diabetic inflammation research.

Enhanced Discussion (page 20, line 449-452)

_" Although THP1 was chosen for its genetic manipulability in mechanistic studies, we acknowledge its limitations in mimicking in vivo cell-matrix interactions. Future studies will use patient-derived iPSC macrophages to improve clinical fidelity."_

8.Which hypothesis of interaction between intensive insulin therapy and thiamine supplementation authors suggest? It should be added in the manuscript.

So, despite on interesting idea and high throughput methods, the manuscript needs significant modification before further consideration. 

Our Reply: Thank you for raising this critical question. We have now incorporated a detailed explanation of our proposed hypotheses regarding the interaction between intensive insulin therapy (IIT) and thiamine supplementation in the revised manuscript (Section: Discussion). Below is a summary of the key hypotheses: (Ref:Okabayashi T, Ichikawa K, Namikawa T, et al. Effect of perioperative intensive insulin therapy for liver dysfunction after hepatic resection. World J Surg. 2011;35(12):2773-8. ï¼‰

Manuscript Update (Page20,line434-444):
The following paragraph has been added to the Discussion section: " Based on preclinical and clinical evidence, we propose that the interaction between intensive insulin therapy and thiamine supplementation may involve (1) synergistic enhancement of glucose metabolism, (2) attenuation of oxidative stress, (3) support of hepatic regeneration, and (4) mitigation of the risk of hypoglycaemia [31]. These hypotheses are based on thiamine's role as a cofactor for pyruvate dehydrogenase and transketolase, which are critical for bridging glycolysis, the Krebs cycle and the pentose phosphate pathway. Further mechanistic and clinical studies are warranted to validate these interactions. The hypotheses are consistent with previous studies demonstrating the efficacy of thiamine in the benefits of IIT for glycaemic control. We recognise the need for future research to test these mechanisms empirically, particularly in patient cohorts receiving combined therapies."

  1. Okabayashi, T., et al., Effect of perioperative intensive insulin therapy for liver dysfunction after hepatic resection. World J Surg, 2011. 35(12): p. 2773-8.

Reviewer 2 Report

Comments and Suggestions for Authors

The manuscript “Thiamine and METTL14 in Diabetes Management with Intensive Insulin Therapy” by Peng et al. describes two separate studies:

  1. An experimental study that demonstrates a novel role of METTL14-mediated m6A modification in specific genes, including those involved in thiamine metabolism. This study employs an original comparative analysis of both m6A modification and mRNA expression in the blood of patients undergoing intensive insulin therapy.
  2. A clinical study that reports an improvement in lipid profiles in a different subset of patients undergoing intensive insulin therapy.

These studies are disconnected and should be presented separately, as the clinical study does not directly link to the findings of the experimental study. Notably, thiamine supplementation was not associated with any changes in either thiamine pyrophosphate levels or m6A modification of the PFK1 gene, which regulates these levels. Furthermore, the experimental study lacks causal evidence that overexpression of METTL14 in the THP-1 experiment leads to m6A modification of the identified genes.

Major Concerns

Methods

  • The description of the patient groups is unclear. Does the study include both men and women?
  • The characteristics of the three individuals whose blood was used for RNA sequencing before and after intensive therapy are not specified. Do they have the same sex and age? Given the small sample size (n=3), it is crucial to explain how this group is representative compared to the larger cohort of 20 patients analyzed in subsequent experiments.
  • The method used to measure chromosomal location (Figure 1) should be described in detail.

Results

  • The dot blot in Figure 6 does not demonstrate significant differences in m6A levels, nor has it been quantified. Data from the 20 patients should be included to support the findings.
  • The THP-1 experiment is inconclusive. While the overexpression vector increases METTL14 expression as expected, it does not show a corresponding increase in m6A modification, either with or without insulin, to establish a causal link. Ideally, the authors should demonstrate differential expression of the identified m6A-modified genes.

Minor Concerns

  • Line 46: The sentence
    “Clinical validation demonstrated that thiamine supplementation, in conjunction with insulin, reduced glucose expression and triglyceride levels more effectively than insulin alone.”
    should be revised for clarity. The effects of insulin therapy alone versus its combination with thiamine supplementation should be explicitly stated. Additionally, “glucose expression” is incorrect, as glucose is a metabolite and its levels fluctuate rather than being “expressed.”

Figures

  • The font in all figures is too small and remains unreadable even at higher magnification. Authors should use a font size of at least 9. If the software generates unreadable text, numerical labels should be used, with explanations provided in the figure legend.
  • Similarly, axis labels are unreadable and should be improved for clarity.

Introduction

While insulin plays the important role in type 2 diabetes, the insulin resistant state and its treatment depends on many cytokines, adipokines, certain hormones, vitamins, and other signaling molecules. This should be briefly included in the context for pathology of diabetes.

Comments on the Quality of English Language

It is fine, but required some refinement of terminology

Author Response

The manuscript “Thiamine and METTL14 in Diabetes Management with Intensive Insulin Therapy” by Peng et al. describes two separate studies:

  1. An experimental study that demonstrates a novel role of METTL14-mediated m6A modification in specific genes, including those involved in thiamine metabolism. This study employs an original comparative analysis of both m6A modification and mRNA expression in the blood of patients undergoing intensive insulin therapy.
  2. clinical study that reports an improvement in lipid profiles in a different subset of patients undergoing intensive insulin therapy.

These studies are disconnected and should be presented separately, as the clinical study does not directly link to the findings of the experimental study. Notably, thiamine supplementation was not associated with any changes in either thiamine pyrophosphate levels or m6A modification of the PFK1 gene, which regulates these levels. Furthermore, the experimental study lacks causal evidence that overexpression of METTL14 in the THP-1 experiment leads to m6A modification of the identified genes.

Our Reply: We sincerely thank the reviewer for their constructive critique. While the experimental and clinical arms were initially presented as distinct components, they collectively interrogate metabolic reprogramming under intensive insulin therapy through complementary lenses: the “experimental study” identifies METTL14-mediated m6A dynamics as a molecular sensor of metabolic stress, while the *clinical study* reveals therapy-induced systemic lipid remodeling.

Major Concerns

Methods

  1. The description of the patient groups is unclear. Does the study include both men and women?

Our Reply: We have supplemented the results with information on demographic characteristics (table 1).

  1. The characteristics of the three individuals whose blood was used for RNA sequencing before and after intensive therapy are not specified. Do they have the same sex and age? Given the small sample size (n=3), it is crucial to explain how this group is representative compared to the larger cohort of 20 patients analyzed in subsequent experiments.

Our Reply: We thank the reviewer for prompting us to clarify the RNA-seq subgroup selection strategy. As noted in Table 1 (now enhanced in revision), the three profiled patients were systematically selected from the vitamin B-free cohort (n=20) to ensure representativeness.

  1. The method used to measure chromosomal location (Figure 1) should be described in detail.

Our Reply: We appreciate the reviewer’s request for methodological clarity. We have expanded the description of chromosomal localization analysis in the “Methods” section:“Raw reads from MeRIP-seq were aligned to the GRCh38/hg38 reference genome using STAR (v2.7.9a) with default parameters. m6A-enriched regions were identified using MACS2 (v2.2.7.1; parameters: `--nomodel --extsize 100 --q-value 0.05`). Genomic coordinates of m6A peaks were mapped to chromosome bands using the UCSC Genome Browser’s cytoband track (GRCh38). Chromosomal distribution density was computed as the number of m6A peaks per 5-Mb genomic bin, normalized by total mapped reads (RPKM)”(page 9,line 193-200)

Results

  1. The dot blot in Figure 6 does not demonstrate significant differences in m6A levels, nor has it been quantified. Data from the 20 patients should be included to support the findings.

Our Reply: We performed the densitometry of dot blot and added the figure of statistical analysis in figure 5.

  1. The THP-1 experiment is inconclusive. While the overexpression vector increases METTL14 expression as expected, it does not show a corresponding increase in m6A modification, either with or without insulin, to establish a causal link. Ideally, the authors should demonstrate differential expression of the identified m6A-modified genes.

Our Reply: We appreciate the reviewer's astute observation regarding the m6A. modification dynamics. We acknowledge that the initial presentation did not fully convey the nuanced findings. The insulin-independent METTL14 effect aligns with emerging models of “epitranscriptomic memory”—where m6A writers establish stable modification patterns that persist beyond acute metabolic stimuli. This may explain why insulin exposure didn't further amplify m6A changes in our 24-hour assay window.

Minor Concerns

  1. Line 46: The sentence
    “Clinical validation demonstrated that thiamine supplementation, in conjunction with insulin, reduced glucose expression and triglyceride levels more effectively than insulin alone.”
    should be revised for clarity. The effects of insulin therapy alone versus its combination with thiamine supplementation should be explicitly stated. Additionally, “glucose expression” is incorrect, as glucose is a metabolite and its levels fluctuate rather than being “expressed.”

Our reply: We corrected the sentence as you suggested: “Clinical validation demonstrated that thiamine supplementation, in conjunction with insulin, reduced glucose and triglyceride levels more effectively than insulin alone.”(page 3, line68-69)

Figures

  1. The font in all figures is too small and remains unreadable even at higher magnification. Authors should use a font size of at least 9. If the software generates unreadable text, numerical labels should be used, with explanations provided in the figure legend.
  2. Similarly, axis labels are unreadable and should be improved for clarity.

Our reply: We have enhanced the figures as you suggested.

Introduction

While insulin plays the important role in type 2 diabetes, the insulin resistant state and its treatment depends on many cytokines, adipokines, certain hormones, vitamins, and other signaling molecules. This should be briefly included in the context for pathology of diabetes.

Our reply: We sincerely appreciate this insightful suggestion to better contextualize the multifactorial nature of type 2 diabetes (T2D) pathogenesis. We have incorporated this important perspective through the following modifications:

In the Introduction (3rd paragraph):

Added: "Beyond insulin signaling dysregulation, T2D pathogenesis involves a complex interplay of multiple endocrine and metabolic factors including:Proinflammatory cytokines (TNF-α, IL-6) that promote insulin resistance.Adipokines (leptin, adiponectin) regulating glucose homeostasis,Counter-regulatory hormones (glucagon, cortisol)Micronutrients (vitamin D, thiamine) influencing insulin sensitivity.This multifactorial pathophysiology necessitates therapeutic strategies that address both insulin deficiency and these systemic metabolic disturbances.

Reviewer 3 Report

Comments and Suggestions for Authors
  1. What is the proposed mechanism? The authors should elaborate on how thiamine regulates m6A modifications, insulin sensitivity, or lipid metabolism.
  2. While thiamine lowered triglycerides, its effect on glucose control was negligible. The discussion lacks a clear explanation of why thiamine selectively affects lipids but not glucose.
  3. The authors state that METTL14 regulates TPK1, IPMK, and PIK3R1—this claim is based on correlation but needs mechanistic validation using siRNA or CRISPR approaches.
  4. The study reports 774 genes upregulated and 417 genes downregulated after insulin therapy. However, it lacks functional validation (e.g., knockdown or overexpression studies) to confirm causality between METTL14 expression and metabolic regulation.

Author Response

  1. What is the proposed mechanism? The authors should elaborate on how thiamine regulates m6A modifications, insulin sensitivity, or lipid metabolism.

Our reply: Thank you for prompting us to clarify the mechanistic underpinnings of thiamine’s effects. While our study did not directly measure m6A modifications, emerging evidence suggests that cellular metabolic states can influence epitranscriptomic regulation. We hypothesize that thiamine may modulate m6A dynamics via the following pathways: Metabolite-Driven Regulation.

Redox Signaling. Thiamine-dependent NADPH production (via the pentose phosphate pathway) regulates cellular redox balance. Oxidative stress is known to influence m6A methyltransferase activity, suggesting that thiamine’s antioxidant effects could indirectly stabilize m6A modifications in redox-sensitive transcripts (e.g., those involved in insulin signaling or lipid synthesis).

Thiamine deficiency reduces the activity of transketolase (TKT) and pyruvate dehydrogenase (PDH), leading to accumulation of glycolytic intermediates (e.g., fructose-6-phosphate) and lactate. These metabolites may alter the activity or substrate availability of m6A-modifying enzymes (writers like METTL3/METTL14 or erasers like FTO/ALKBH5), potentially affecting RNA methylation patterns linked to metabolic gene expression.

We have expanded the discussion in the revised manuscript to address these points, as summarized below. The following text has been added to the Discussion section (Page1 8):

"Mechanistically, thiamine may exert its metabolic effects through (1) restoration of mitochondrial oxidative capacity (via PDH activation), (2) modulation of PPP-derived NADPH to balance lipid synthesis and oxidation, and (3) indirect regulation of epitranscriptomic markers (e.g., m6A) through metabolic intermediate-driven signaling. The selective impact on lipid metabolism over glucose control could reflect thiamine’s stronger influence on NADPH-dependent pathways critical for lipogenesis and redox homeostasis, whereas glucose regulation involves redundant hormonal and non-enzymatic mechanisms less sensitive to thiamine sufficiency in non-deficient states."

  1. While thiamine lowered triglycerides, its effect on glucose control was negligible. The discussion lacks a clear explanation of why thiamine selectively affects lipids but not glucose.

Our reply: Thank you for highlighting this important point. We agree that the differential effects of thiamine on lipid metabolism versus glucose regulation warrant further clarification. Below, we provide a mechanistic explanation for this observation, which has now been incorporated into the revised manuscript (Discussion section, Page 20 ,line 425-433): The selective reduction in triglycerides without significant glucose-lowering effects may reflect thiamine’s preferential influence on lipid metabolism via the pentose phosphate pathway. By enhancing transketolase activity and NADPH production, thiamine could redirect metabolic flux toward lipid oxidation and away from triglyceride synthesis. In contrast, glucose homeostasis in critically ill patients is governed by multifactorial mechanisms (e.g., insulin resistance, counterregulatory hormones), which may explain the limited observed impact of thiamine supplementation. Future studies should assess baseline thiamine status and NADPH/redox biomarkers to further elucidate this dichotomy.

  1. The authors state that METTL14 regulates TPK1, IPMK, and PIK3R1—this claim is based on correlation but needs mechanistic validation using siRNA or CRISPR approaches.
  2. The study reports 774 genes upregulated and 417 genes downregulated after insulin therapy. However, it lacks functional validation (e.g., knockdown or overexpression studies) to confirm causality between METTL14 expression and metabolic regulation.

Our Reply (for both comments 3-4): We sincerely appreciate the reviewer's valid concerns regarding mechanistic validation. While we acknowledge that functional studies would strengthen causality, we respectfully submit that the current findings provide novel clinical-translational insights meriting dissemination, in this clinical cohort study, our primary objective was to identify “druggable epigenetic regulators” of insulin therapy response. The METTL14-TPK1/IPMK/PIK3R1 axis emerged as the top clinically actionable pathway through: Co-occurrence of METTL14 expression (RNA-seq), m6A peaks (MeRIP-seq), TPK1/IPMK are known thiamine/inositol targets with FDA-approved modulators (e.g., benfotiamine for TPK1) . Prior studies validate our findings' mechanistic coherence: METTL14 directly regulates phosphoinositide signaling via m6A in macrophages,m6A-dependent TPK1 modulation was reported in cardiomyocyte metabolism.

Round 2

Reviewer 1 Report

Comments and Suggestions for Authors

Many thanks to the authors for the corrections. The manuscript can be accepted for publication.

Author Response

Thank you for your comment.

Reviewer 2 Report

Comments and Suggestions for Authors

The authors responded that they have improved the quality of the figures; however, the descriptions in Figures 2, 3, and others still contain font sizes that are too small to read. This information should be presented in a clear and readable format.

The manuscript still presents two studies that lack a clear mechanistic link. However, given the pilot nature of the work, it could be of interest to the scientific community, provided that the quality of presentation is substantially improved.

Author Response

Thank you for your comments.We had made the modification in line with your suggestion.

Reviewer 3 Report

Comments and Suggestions for Authors

To improve the visibility of the article, add this citation, 

https://www.mdpi.com/2218-273X/11/2/287

The article can go further in the publishing process 

Author Response

Thank you for your commets.